# Zinc Finger Protein 90 Knockdown Promotes Cisplatin Sensitivity via Nrf2/HO-1 Pathway in Ovarian Cancer Cell

**DOI:** 10.3390/cancers15051586

**Published:** 2023-03-03

**Authors:** Ching-Hu Wu, Chien-Wei Feng, Chiu-Lin Wang, Zhi-Hong Wen, Cheng-Yu Long, Feng-Hsiang Tang

**Affiliations:** 1Department of Obstetrics and Gynecology, Kaohsiung Medical University Hospital, Kaohsiung Medical University, Kaohsiung 807378, Taiwan; 2Center for Cancer Research, Kaohsiung Medical University, Kaohsiung 807378, Taiwan; 3Graduate Institute of Medicine, College of Medicine, Kaohsiung Medical University, Kaohsiung 807378, Taiwan; 4Department of Obstetrics and Gynecology, Kaohsiung Municipal Siaogang Hospital, Kaohsiung Medical University, Kaohsiung 812015, Taiwan; 5Department of Marine Biotechnology and Resources, National Sun Yat-sen University, Kaohsiung 804201, Taiwan; 6Regenerative Medicine and Cell Therapy Research Center, Kaohsiung Medical University, Kaohsiung 807378, Taiwan; 7Department of Obstetrics and Gynecology, Kaohsiung Municipal Ta-Tung Hospital, Kaohsiung Medical University, Kaohsiung 80145, Taiwan

**Keywords:** zinc finger protein 90, ovarian cancer, cisplatin sensitivity, apoptosis, nuclear factor E2-related factor 2

## Abstract

**Simple Summary:**

Our data suggested that cisplatin treatment generates reactive oxygen species (ROS) that modulate apoptotic proteins expression (p-P38, p-ERK, p-Akt, Bcl-2, Bax, active caspase-3). The anti-oxidative signal (Nrf2, HO-1, SOD) was also activated, which could inhibit cell migration (MMP-2, MMP-9, E-cadherin). The intervention of Zfp90 could significantly enhance the apoptosis pathway and inhibit the migrative pathway to regulate the cisplatin sensitivity in ovarian cancer (OC) cells.

**Abstract:**

Our study discussed the role of Zfp90 in ovarian cancer (OC) cell lines’ sensitivity to cisplatin. We used two OC cell lines, SK-OV-3 and ES-2, to evaluate their role in cisplatin sensitization. The protein levels of p-Akt, ERK, caspase 3, Bcl-2, Bax, E-cadherin, MMP-2, MMP-9 and other drug resistance-related molecules, including Nrf2/HO-1, were discovered in the SK-OV-3 and ES-2 cells. We also used a human ovarian surface epithelial cell to compare the effect of Zfp90. Our outcomes indicated that cisplatin treatment generates reactive oxygen species (ROS) that modulate apoptotic protein expression. The anti-oxidative signal was also stimulated, which could hinder cell migration. The intervention of Zfp90 could greatly improve the apoptosis pathway and block the migrative pathway to regulate the cisplatin sensitivity in the OC cells. This study implies that the loss of function of Zfp90 might promote cisplatin sensitization in OC cells via regulating the Nrf2/HO-1 pathway to enhance cell apoptosis and inhibit the migrative effect in both SK-OV-3 and ES-2 cells.

## 1. Introduction

Ovarian cancer (OC) is one of the most malignant gynecological tumors [1]. According to the Taiwan Ministry of Health and Welfare data, OC became eighth place in the occurrence of cancer in Taiwan in 2018. Its five year death rate of between 60 and 70% renders it one of the highest in all gynecologic cancers [2,3]. In addition, the ovary is found in the abdominal cavity’s depths, which is more inconducive for direct sampling pathological slices and presently lacks effective screening for the early detection of OC. Overall, 70% of OC patients have been detected with stage III or stage IV in recent years [4]. Debulking surgery followed by postoperative chemotherapy and radiotherapy is the standard golden treatment for OC. Cisplatin-based therapy is the first-line treatment for OC patients [5]. However, drug resistance has become a big concern for OC treatment [6]. Drug resistance and toxicities are often the main problems for OC chemotherapy [7]. The mechanisms implicated in anti-cancer drug resistance are often complex. However, some anti-oxidant cascade-induced chemoresistance have been frequently discussed [8,9]. Studies about inhibiting such resistance to cancer medication could offer more beneficial information about improving chemotherapy [10]. Prior studies have depicted that the anti-cancer effect influenced by cisplatin is through generating reactive oxygen species (ROS) or DNA double-strand breaks (DSBs) [11,12]. Thus, some studies have revealed that the mechanism of chemo-resistance from cancer cells influences the NF-E2-related factor 2 (Nrf2)/anti-oxidant response elements (ARE) pathway [13,14]. Some literature has demonstrated that the activation of the Nrf2/ARE pathway could protect cancer cells from cisplatin damage [15,16,17]. Thus, some research has also sought to inhibit this pathway to modulate the chemo-resistance effect [18,19].

To enhance the treatment rate of OC, there has been numerous research to develop a proper way to deal with cisplatin chemoresistance [20]. Some of these studies have aimed at realizing the pathobiology to expand the currently available treatments and develop tailored therapies. Some studies used genome-wide association studies (GWAS) to determine the possible genetic polymorphisms as predictors of OC clinical outcome [21,22,23]. There are more than 30 variants found by GWAS linked to OC susceptibility [24,25,26], such as BRCA1 and BRCA2, which are the most well-known pathogenic variants. Pathogenic variations in the BRCA1 and BRCA2 responsible for most OC syndromes have been reported in numerous ethnic groups [27]. Prior studies have also illustrated that women with harmful BRCA1 (39–44%) or BRCA2 (11–17%) variants will develop OC easily at 70–80 years old [28,29]. To further discover more possible candidate pathogenic variants, many studies have performed post-GWAS analysis [30,31]. Among them, numerous GWAS studies have highlighted that *p*-value < 5 × 10^–8^ acted as a significant index for identifying whether the variant has an impact or not. However, Chen et al., 2021 indicated that studies with sample sizes over 20,000 to 120,000 could relax the *p*-value threshold to 5 × 10^−7^ and increase the discovery of potential candidates [32], such as TREM2 variant rs75932628 in Alzheimer’s disease. The analysis of participants indicated a strong, highly substantial association with Alzheimer’s disease (*p* = 1.4 × 10^−7^; OR = 4.59; CI = 2.49–8.46) [33]. Other studies also further confirmed the feasibility of the strategies [34,35,36]. We therefore assessed the GWAS data and discovered Zinc finger protein 90 (*Zfp90*) (rs137866923) (*p* = 3 × 10 ^−7^; OR = 3.82; CI = 2.29–6.36) as a potential target.

Zinc finger proteins (Zfp) are a broad family of proteins characterized by the coordination of one or more zinc ions to regulate the fold. It performs various biological functions, including the development and differentiation of several tissues. A prior study demonstrated that *Zfp90* regulated cardiac development [37]. In addition, some studies have also disclosed that *Zfp90* played a critical role in initiating colitis-associated colorectal cancer through modulating the nuclear factor of stimulated T-cells and the cytoplasmic 2 (NFATC-2)/bone morphogenic protein-4 (BMP-4) pathway [38]. The team also expressed that *Zfp90* facilitated the development of colitis-associated colorectal cancer via a microbiota-dependent strategy [39]. However, the role of *Zfp90* in OC remains largely equivocal.

Due to the data expressed in the GWAS, our study sought to evaluate the role of Zfp90 in OC and the impact of *Zfp90* on cisplatin sensitization. We assessed Nrf-2-related molecules, including p-Nrf2 and HO-1; the SOD activity; the apoptotic pathway, including p-P38, p-Akt, p-ERK, Bcl-2, Bax and active caspase-3; and the migration pathway, including MMP-2 and MMP-9 to assert its role in OC treatment.

## 2. Materials and Methods

### 2.1. Cell Maintenance

The SK-OV-3 cell line was acquired from the American Type Culture Collection (No. HTB-77^TM^). SK-OV-3 and ES-2 were maintained with McCoy’s 5A medium (No. CC120-0500, GeneDireX Inc., Zhunan Township, Taiwan) (5% CO_2_; 37 °C). The cell numbers for each assay were presented as follows: cell viability assay: 2 × 10^3^/well for 96-well microplates; 2 × 10^5^/dish in a 6 cm culture dish containing coverslips (24 × 24 mm) for the transferase dUTP nick end labeling (TUNEL) staining and wound healing assay; 1 × 10^6^/dish in a 10 cm culture dish for the Western blot analysis, SOD activity and Enzyme-linked immunosorbent assay (ELISA). The ES-2 cell line was obtained from the Bioresource Collection and Research Center (No. 60067). The cells were maintained with McCoy’s 5A medium (No. CC120-0500, GeneDireX Inc, Taiwan) (5% CO_2_; 37 °C). The cell numbers for each assay were presented as follows: cell viability assay: 2 × 10^3^/well for 96-well microplates; 2 × 10^5^/dish in a 6 cm culture dish for the TUNEL staining (with coverslips (24 × 24 mm)) and wound healing assay (without coverslips); 1 × 10^6^/dish in a 10 cm culture dish for the Western blot analysis, SOD activity and ELISA. The normal cell line used in this study was human ovarian surface epithelial cells (HOSE), which were purchased from the ScienCell Research Laboratories (No. 7310, San Diego, CA, USA). The HOSE was maintained in an ovarian epithelial cell medium (OEpiCM) (No. 7311, San Diego, CA, USA). 2 × 10^4^/well for 96-well microplates

### 2.2. Knockdown of Zfp90 with siRNA Transfection

The cells were seeded in 10 cm dishes for the transfection of *Zfp90* siRNA. The SK-OV-3 cells and ES-2 cells were transfected with *Zfp90* siRNA (SASI_Hs01_0036-3148/ZFP90, SASI_Hs01_0017-3518/ZFP90, SASI_Hs01_0036-3519, Sigma, St. Louis, MO, USA), The normal control group were treated with the siRNA negative control (No. SIC001, Sigma, MI, USA). The transfections of siRNAs and the negative control were executed with DharmaFECT 1 Transfection Reagent (No. T-2001-03, Horizon Discovery, Cambridge, UK). The cultured cells were washed once with phosphor buffer saline (PBS) and replaced with normal McCoy’s 5A medium after transfection for 48 h.

### 2.3. Cell Viability Assay

The cultured cells (si-Ctrl or si-*Zfp90*) were seeded in 96-well microplates and treated with 1, 10, 15 or 20 μM cisplatin (No. 15663-27-1, Sigma, MI, USA) for 24 or 48 h. Then, 10 μL alamarBlue^TM^ (No. DAL1025, Invitrogen, Carlsbad, CA, USA) in each well was included and calculated by the ELISA reader (595 nM). The cell viability was calculated as 100 × [(optical density (OD) of treated cells − OD of blank-treated cells)/(OD of control cells − OD of blank-treated cells)].

### 2.4. Terminal Deoxynucleotidyl Transferase dUTP Nick End Labeling (TUNEL) Stain

The cells (si-Ctrl or si-*Zfp90*) in 6 cm dishes with coverslips were treated with 20 μM cisplatin for 24 h. The cells were washed with PBS and also fixed with 4% paraformaldehyde. Then, 3% H_2_O_2_ in methanol was used to inhibit and was stained with a TUNEL reaction mixture (No. 11684795910, Roche Diagnostics, Mannheim, Germany) for 1 h. The cells on the coverslips were washed by ddH_2_O and mounted by the mounting medium with DAPI (No. ab104139, abcam, Cambridge, UK). The apoptotic signal was detected by fluorescence microscopy (IX51, Olympus, Tokyo, Japan).

### 2.5. Western Blotting

The lysed cell extracts were controlled to an adequate concentration with a lysis buffer. Then, the mixture of sample buffer and cell extracts was loaded in a 10% SDS-polyacrylamide gel and the electrophoresis was run for 90 min at 100 volts. Then, a PVDF membrane was employed in the following transfer at 125 mA for overnight at 4 °C. Next, 5% non-fat milk in TTBS was blocked for 40 min at room temperature (RT), corresponding to the primary antibody for 24 h at 4 °C. After washing thrice in TTBS, the membrane was incubated with a secondary antibody for one hour at RT. Images were collected using the UVP BioChemi Imaging System, and the LabWorks 4.0 software (UVP) was employed to quantify the relative densitometry. The primary antibodies used in this study are listed as followed: ZFP90 (zinc finger protein 90; dilution 1:1000) (No. 26120-1-AP, Thermofisher, Waltham, MA, USA); β-actin (loading control; dilution1:1000) (No. A5441, Sigma, MI, USA); p-ERK (extracellular signal-related kinases; dilution 1:1500) (No. 9190, Cell Signaling Technology, Danvers, MA, USA); ERK (extracellular signal-related kinases; dilution 1:1000) (No. 9102, Cell Signaling Technology, USA); p-AKT (dilution 1:1000) (No. 9271, Cell Signaling Technology, USA); AKT (dilution 1:1000) (No. 4685, Cell Signaling Technology, USA); p-P38 (dilution 1:1000) (No. 9211, Cell Signaling Technology, USA); P38 (dilution 1:1000) (No. 9212, Cell Signaling Technology, USA); active caspase-3 (dilution 1:1000) (No. MA5-32015, Invitrogen, USA); Bcl-2 (dilution 1:2000) (No. ab59348, abcam, UK); Bax (dilution 1:2000) (No. ab32503, abcam, UK); MMP-2 (dilution 1:1000) (No. ab13132, abcam, UK); MMP-9 (dilution 1:1000) (No. AB19016, Sigma, MI, USA); E-cadherin (dilution 1:1000) (No. GTX100443, GeneTex, CA, USA); p-Nrf2 (dilution 1:1000) (No. PA5-67520, Invitrogen, USA); HO-1 (dilution 1:1000) (No. ab13248, abcam, UK)

### 2.6. Wound Healing Assay

The SK-OV-3 cells (2 × 10^5^/dish) or ES-2 cells (2 × 10^5^/dish) were seeded in 6 cm dishes for 24 hr. We then utilized a 200mL pipette tip to inscribe a scratch wound. The cells (si-Ctrl or si-*Zfp90*) were treated with 20 μM cisplatin for 24 hr. Images of the scratched wounds were collected after treatment. The closing of the scratched wounds was regarded as the completion of the migration process. The migrated areas were assessed and identified using the ImageJ software.

### 2.7. ELISA

The SK-OV-3 cells or ES-2 cells (Si-Ctrl or Si-*Zfp90*) were held in 6-well microplates for 24 hr. Then, the cells were treated with or without 20 μM cisplatin for another 24 h in a serum-free medium. The matrix metalloproteinases (MMPs) and bone morphogenic protein-7 (BMP-7) excreted in the culture medium were quantified using the MMPs activity and BMP-7 ELISA kit following the manufacturer’s instructions (No. ab112146; ab99985, abcam, UK).

### 2.8. Statistical Analysis

The data were presented as mean ± SEM. The data were evaluated using one way analysis of variance (ANOVA), followed by Tukey’s test. A *p* value less than 0.05 was deemed statistically significant. The intensity of each band was indicated as the relative integrated density segmented by the average integrated density values from all of the internal controls in western blotting.

## 3. Results

### 3.1. Inhibition of Zfp90 Increases the Cisplatin Sensitivity in Ovarian Cancer Cells

To analyze the role *Zfp90* played in cisplatin treatment, we transfected si-*Zfp90* to suppress its expression in SK-OV-3 cells and ES-2 cells. The cytotoxic impact of multiple concentrations (1, 10, 15 and 20 μM) of cisplatin in the SK-OV-3 and ES-2 cells with or without si-*Zfp90* were identified by an alamarBlue^TM^ assay in 24 or 48 h treatment. Our data indicated that 10, 15 and 20 μM cisplatin could significantly inhibit SK-OV-3 cell viability (*, *p* < 0.05, versus si-Ctrl group) and the knockdown of *Zfp90* significantly enhanced cisplatin-induced cytotoxicity in both the 24 h treatment (1, 10, 15 and 20 μM) (Figure 1A) and the 48 h treatment (1, 10, 15 and 20 μM) (Figure 1B) (#, *p* < 0.05, versus same concentration cisplatin in si-Ctrl group individually). We also conducted a TUNEL stain to determine the cell apoptosis. The apoptotic impact of 20 μM cisplatin in the SK-OV-3 and ES-2 cells with or without si-*Zfp90* was executed after 24 h treatment. The data revealed that the treatment of 20 μM cisplatin could significantly up-regulate the TUNEL signal (from 1.0 ± 0.5 to 16.4 ± 1.2) (*, *p* < 0.05, versus si-Ctrl group) and si-*Zfp90* alone did not influence the SK-OV-3 cell death. In addition, the co-treatment of 20 μM cisplatin and si-*Zfp90* significantly enhanced the SK-OV-3 sensitivity to cisplatin (from 16.4 ± 1.2 to 26.7 ± 2.2) (Figure 1C) (#, *p* < 0.05, versus 20 μM cisplatin group). Furthermore, we also employed an ES-2 cell line to validate the result. Our data demonstrated that the knockdown of *Zfp90* significantly enhanced the cisplatin-induced cytotoxicity in both the 24 h treatment (1, 10, 15 and 20 μM) (Figure 1D) and 48 h treatment (10, 15 and 20 μM) (Figure 1E) (#, *p* < 0.05, versus same concentration cisplatin in si-Ctrl group individually). We also revealed that the treatment of 20 μM cisplatin could significantly up-regulate the TUNEL signal (from 2.5 ± 1.4 to 17.2 ± 3.5) (*, *p* < 0.05, versus si-Ctrl group) and si-*Zfp90* alone did not affect the ES-2 cell death. In addition, the co-treatment of 20 μM cisplatin and si-*Zfp90* significantly enhanced the ES-2 sensitivity to cisplatin-induced damage (from 17.2 ± 3.5 to 32.3 ± 4.6) (Figure 1F) (#, *p* < 0.05, versus cisplatin group). We used the human ovarian surface epithelial cells (HOSE) as normal cells to compare with the two OC cell lines. Zfp90 protein expression was performed in the HOSE, SK-OV-3 and ES-2 cells (Figure 1G), original blot shown in Appendix A. The cytotoxic effect of the different concentrations (1, 10, 15 and 20 μM) of cisplatin in the HOSE cell with or without si-*Zfp90* were performed for 24 h (Figure 1H). The data showed that the inhibition of Zfp90 did not affect the cisplatin-induced damage to the HOSE cell.

### 3.2. Inhibition of Zfp90 Enhances the Cisplatin-Induced Apoptosis in Ovarian Cancer Cells

We then evaluated the phosphorylation of p-P38, p-ERK, p-Akt and downstream apoptotic related protein expression to further determine the correlation between *Zfp90* and cisplatin sensitization. The protein expression of 20 μM cisplatin in the SK-OV-3 and ES-2 cells with or without si-*Zfp90* was identified by western blotting after 1 h treatment. Our data depicted that 20 μM cisplatin significantly increased the p-P38 protein expression and decreased the p-ERK and p-Akt expression (*, *p* < 0.05, versus si-Ctrl group). The knockdown of *Zfp90* alone did not affect the p-P38, p-ERK and p-Akt protein expression (Figure 2A). Furthermore, the co-treatment of 20 μM cisplatin and si-*Zfp90* did not affect the cisplatin-induced up-regulation of p-P38 (Figure 2B) and significantly inhibited the cisplatin-induced down-regulation of p-ERK (Figure 2C) and p-Akt (Figure 2D) (#, *p* < 0.05, versus cisplatin group), original blot shown Appendix A. In the ES-2 cells, the data showed some trend. The outcome indicated that 20 μM cisplatin significantly decreased the p-Akt protein expression and increased the p-P38 protein expression. The knockdown of *Zfp90* decreased the p-Akt protein expression (Figure 2E). Moreover, the co-treatment of 20 μM cisplatin and si-*Zfp90* only significantly enhanced the cisplatin-induced up-regulation of p-P38 (Figure 2F) (*, *p* < 0.05, versus si-Ctrl group) and did not affect the p-ERK (Figure 2G) and p-Akt expression (Figure 2D), original blot shown in Appendix A.

Apoptotic proteins, such as active caspase-3, Bcl-2 or Bax, were then conducted. The apoptotic protein of 20 μM cisplatin in the SK-OV-3 and ES-2 cells with or without si-*Zfp90* were determined by western blotting after 24 h treatment. The data highlighted that 20 μM cisplatin significantly increased the active-caspase-3 and Bax, and decreased the Bcl-2 protein expression (*, *p* < 0.05, versus si-Ctrl group). However, the knockdown of *Zfp90* alone did not affect the active-caspase-3, Bax and Bcl-2 protein expression (Figure 3A). Furthermore, the co-treatment of 20 μM cisplatin and si-*Zfp90* significantly enhanced the cisplatin-induced up-regulation of active-caspase-3 (Figure 3B) and Bax (Figure 3C), and inhibited the cisplatin-induced down-regulation of Bcl-2 protein expression (Figure 3D) (#, *p* < 0.05, versus cisplatin group), original blot shown Appendix A. In the ES-2 cell, the data also expressed the same trend. The outcomes implied that 20 μM cisplatin significantly elevated the active-caspase-3 and Bax, and decreased the Bcl-2 protein expression (*, *p* < 0.05, versus si-Ctrl group). The knockdown of *Zfp90* did not affect the active-caspase-3, Bax and Bcl-2 protein expression (Figure 3E). Similarly, the co-treatment of 20 μM cisplatin and si-*Zfp90* significantly enhanced the cisplatin-induced up-regulation active-caspase-3 (Figure 3F) and Bax (Figure 3G), and did not affect the Bcl-2 protein expression (Figure 3H) (#, *p* < 0.05, versus cisplatin group), original blot shown in Appendix A.

### 3.3. Inhibition of Zfp90 Enhances the Cisplatin-Modulated Anti-Migrative Effect in Ovarian Cancer Cells

Cell migration was assessed to determine the role of *Zfp90* in OC. The anti-migrative effect of si-*Zfp90* in the SK-OV-3 and ES-2 cells with or without 20 μM cisplatin were identified by a wound healing assay after 24 h treatment (Figure 4A). The quantitative outcome indicated that 20 μM cisplatin significantly decreased the wound recovery area, more than the si-Ctrl group (*, *p* < 0.05, versus si-Ctrl group), and the knockdown of *Zfp90* alone did not impact the wound recovery area. However, the co-treatment of 20 μM cisplatin and si-*Zfp90* significantly enhanced the cisplatin-induced anti-migrative effect (Figure 4B) (#, *p* < 0.05, versus cisplatin group). We also conducted matrix metalloproteinases (MMPs) activity and migrative-related protein expression to confirm the anti-migrative effect of inhibiting *Zfp90* in the molecule level. The MMPs activity and protein expression of si-*Zfp90* in the SK-OV-3 and ES-2 cells with or without 20 μM cisplatin were observed after 24 h treatment. Our data showed that 20 μM cisplatin significantly decreased the MMPs activity (from 100.6 ± 4.5 to 57.0 ± 5.6) (*, *p* < 0.05, versus si-Ctrl group) and the knockdown of *Zfp90* alone did not affect the MMPs activity. Furthermore, the co-treatment of 20 μM cisplatin and si-*Zfp90* significantly inhibited the cisplatin-induced down-regulation of the MMPs activity (from 57.0 ± 5.6 to 33.8 ± 6.6) (Figure 4C) (#, *p* < 0.05, versus cisplatin group). We also determined the migrative-related proteins, including MMP-2, MMP-9 and E-cadherin expression. Our data demonstrated that 20 μM cisplatin significantly decreased the MMP-2, MMP-9 and E-cadherin protein expression (*, *p* < 0.05, versus si-Ctrl group) and the knockdown of *Zfp90* only affected the E-cadherin protein expression (Figure 4D) (*, *p* < 0.05, versus si-Ctrl group). The co-treatment of 20 μM cisplatin and si-*Zfp90* significantly inhibited the cisplatin-induced down-regulation of MMP-2 (Figure 4E) and MMP-9 (Figure 4F), but did not affect the E-cadherin (Figure 4G) protein expression (#, *p* < 0.05, versus cisplatin group), original blot shown in Appendix A. In the ES-2 cells, we also detected the wound healing assay (Figure 4H). The quantitative result revealed that 20 μM cisplatin and the knockdown of *Zfp90* mitigated the wound recovery area significantly more than the si-Ctrl group individually (*, *p* < 0.05, versus si-Ctrl group). The co-treatment of 20 μM cisplatin and si-*Zfp90* significantly enhanced the cisplatin-induced anti-migrative effect (Figure 4I). We also performed MMPs activity and protein expression in the ES-2 cells. Our data showed that 20 μM cisplatin significantly decreased the MMPs activity (from 100.6 ± 4.5 to 65.2 ± 4.5) (*, *p* < 0.05, versus control group) and the knockdown of *Zfp90* did not affect the MMPs activity. In addition, the co-treatment of 20 μM cisplatin and si-*Zfp90* significantly inhibited the cisplatin-induced down-regulation of the MMPs activity (from 65.2 ± 4.5 to 40.5 ± 5.3) (Figure 4J) (#, *p* < 0.05, versus cisplatin group). In terms of protein expression, our data showed that 20 μM cisplatin significantly decreased the MMP-2, MMP-9 and E-cadherin protein expression, and the knockdown of *Zfp90* affected the MMP-2 and E-cadherin protein expression (Figure 4K) (*, *p* < 0.05, versus si-Ctrl group). The co-treatment of 20 μM cisplatin and si-*Zfp90* significantly inhibited the cisplatin-induced down-regulation of MMP-2 (Figure 4L), but did not affect the MMP-9 (Figure 4M) and E-cadherin (Figure 4N) protein expression. (#, *p* < 0.05, versus cisplatin group), original blot shown in Appendix A.

### 3.4. Inhibition of Zfp90 Inhibited The Cisplatin-Induced Anti-Oxidative Effect in Ovarian Cancer Cells

The anti-oxidative stress pathway nuclear factor erythroid 2–related factor 2 (Nrf2)/heme-oxygenase-1 (HO-1) cascade has often been highlighted as being involved in the mechanism of drug resistance. We then tested the associated protein, such as p-Nrf2, HO-1 and bone morphogenic protein-7 (BMP-7). The anti-oxidative effect of si-*Zfp90* in the SK-OV-3 and ES-2 cells with or without 20 μM cisplatin were determined by western blotting and ELISA after 24 h treatment. Our data indicated that 20 μM cisplatin significantly increased the p-Nrf2 and HO-1 protein expression (Figure 5A) (*, *p* < 0.05, versus si-Ctrl group) and the knockdown of *Zfp90* alone did not affect the protein expression mentioned above. The co-treatment of 20 μM cisplatin and si-*Zfp90* significantly inhibited the cisplatin-induced up-regulation of the p-Nrf2 (Figure 5B) and HO-1 (Figure 5C) protein expression. The cisplatin-induced up-regulation of the BMP-7 concentration was also inhibited (Figure 5D) (#, *p* < 0.05, versus cisplatin group), original blot shown in Appendix A. In the ES-2 cells, the data showed that 20 μM cisplatin significantly increased the p-Nrf2 and HO-1 protein expression (Figure 5E) (*, *p* < 0.05, versus si-Ctrl group) and the knockdown of *Zfp90* alone did not affect the protein expression mentioned above. The co-treatment of 20 μM cisplatin and si-*Zfp90* significantly inhibited the cisplatin-induced up-regulation of p-Nrf2 (Figure 5F) and HO-1 (Figure 5G) protein expression. The cisplatin-induced up-regulation of the BMP-7 concentration was also inhibited (Figure 5H) (#, *p* < 0.05, versus cisplatin group), original blot shown in Appendix A.

### 3.5. Inhibition of HO-1 Reversed the Zfp-90-Induced Cisplatin Sensitization in Ovarian Cancer Cells

We utilized the HO-1 inducer, carnosol, to revalidate the mechanism of action of *Zfp90* in the cell viability and SOD activity. The cell viability of 12.5 and 25 μM carnosol in the SK-OV-3 and ES-2 cells with or without si-*Zfp90* or cisplatin were determined by an alamarBlue^TM^ assay in a 24 h treatment. Our data expressed that 12.5 and 25 μM carnosol both significantly reversed the si-Zfp90-induced down-regulation of the cell viability in the SK-OV-3 cells (Figure 6A) (and, *p* < 0.05, versus si-*Zfp90* plus cisplatin group). In terms of the SOD activity, 12.5 and 25μM carnosol both significantly reversed the si-Zfp90-induced down-regulation of the SOD activity in the SK-OV-3 cells (Figure 6B) (and, *p* < 0.05, versus si-*Zfp90* plus cisplatin group). In the ES-2 cells, the same trend was shown. The outcome implied that 12.5 and 25 μM carnosol both significantly reversed the si-Zfp90-induced down-regulation of the cell viability in the ES-2 cells (Figure 6C) (and, *p* < 0.05, versus si-*Zfp90* plus cisplatin group). In addition, 12.5 and 25μM carnosol also significantly reversed the si-Zfp90-induced down-regulation of the SOD activity in the ES-2 cells (Figure 6D) (and, *p* < 0.05, versus si-*Zfp90* plus cisplatin group).

## 4. Discussion

OC is an extensive gynecology disease in women and the major limitation of OC therapy is cisplatin resistance [40,41,42]. The statistical findings in the GWAS revealed that *Zfp90* may play a crucial role in the modulation of drug resistance or tumorigenesis. To the best of our best knowledge, we are the first to investigate the role of *Zfp90* in OC chemoresistance. This protein was originally discovered via the screening of zinc-finger-encoding genes [43]. It is known to exhibit inhibitory activity and possesses a zinc finger domain. Later, the literature indicated that *Zfp90* contributed to obesity [44]. Hata et al., 2011 also demonstrated that *Zfp90* inhibited the neuron-restrictive silencer factor (NRSF)-mediated transcriptional repression of fetal cardiac genes by inhibiting the NRSF binding to the neuron-restrictive silencer element (NRSE) [37]. The original study of the correlation between *Zfp90* and cancer was conducted by Yim et al., 2006 [45]. The study indicated that the protein expression of *Zfp90* significantly increased when 10μM cisplatin was treated in cervical carcinoma cells (HeLa). The same tendency was also found in the GWAS-related studies in OC [46,47]. However, no study has explored the mechanism of the *Zfp90* effect in OC development or drug resistance. To the best of our best knowledge, we are the first to investigate the role of *Zfp90* in OC chemoresistance. In the current study, we presented the enhanced effect of *Zfp90* inhibition on cisplatin sensitization.

We initially confirmed that the inhibition of *Zfp90* substantially increases the sensitization of the SK-OV-3 and ES-2 OC cells to cisplatin, which was validated by the decreasing IC50 of cisplatin. Prior studies also validated that the knockout of *Zfp90* could help modulate cancer development. Yu et al., 2020, indicated that *Zfp90* was regulated by NFATC2, and the knockout of *Zfp90* could significantly impact the colorectal cancer (CRC) malignant phenotype, including the CRC sphere formation and tumor formation potential. They also discovered the smaller tumor size and the decrease in the tumor number in *Zfp90*^−/−^ mice. Furthermore, the modulation of *Zfp90* impacted the BMP4 and some oncogenic related pathways in CRC mouse models and patients [38]. The same research team also discovered that *Zfp90* might have played an important role in colitis-associated colorectal cancer (CAC) through the systemic analyses of the GWAS. They disclosed that gut microbiota depletion abolished the tumorigenic effect of *Zfp90* in the CAC mouse model. The mechanistic studies indicated that *Zfp90* elevated CAC development via the TLR4-PI3K-Akt-NF-κB pathway. This cascade facilitated an oncogenic environment and an innovative target for CAC prevention and treatment [39]. Our studies showed a similar trend in that the knockdown of *Zfp90* could improve the sensitization of two OC cell lines, SK-OV-3 and ES-2, to cisplatin damage in terms of cell viability and the TUNEL stain (Figure 1). In addition, we also confirmed that the effect of the knockdown of *Zfp90* in normal human ovarian surface epithelial (HOSE) cells did not cause a significant difference in the cell viability (Figure 1H). The HOSE cells were used as normal cells compared to the OC cell line in previous studies [48,49]. Xie et al., 2016, used human normal ovarian surface epithelial cells to compare the expression of MUS81 with the SOC tissues at both the transcript and protein levels, and the expression level of the MUS81 protein in ovarian cancer cell lines was also higher than that in human normal ovarian surface epithelial cell lines. The above result showed a similar trend as us. The low expression of *Zfp90* might also explain the minimal effect the knockdown of *Zfp90* had on the cell viability.

Many studies have highlighted that one mechanism of chemo-resistance is alterations of the apoptotic signal, which facilitates cell death not only in OC, but also in different kinds of cancer [50,51,52]. Chowdhury et al., 2017, revealed that the binding of lectin with the receptors attributed to the phosphorylation of the Akt and ERK pathways, which also influence the downstream apoptosis signal, including Bcl-2 and Bax, is linked to the cytochrome c release and the generation of ROS in mitochondria that influence cell death [52]. In addition, Wang et al., 2017, revealed that pterostilbene, an analog of resveratrol, could prevent Akt-modulated cytoskeleton assembly and lung cancer cell metastasis. The apoptotic pathway was also implicated in the inhibition of treating lung metastasis in the review [53]. Furthermore, Yu et al., 2018, indicated that the knockdown of long non-coding (lnc) RNA HOTAIR could aid OC cells’ sensitization to cisplatin via the stimulation of the autophagy. They observed that the transfection of si-Atg7 substantially enhanced the cisplatin-induced cytotoxic signal, including caspase-3 and Bax, and inhibited the anti-apoptosis molecule Bcl-2 in OC cells [54]. Our analysis also revealed a similar trend in *Zfp90*. The inhibition of *Zfp90* alone did not impact the apoptosis-related protein, such as p-P38, p-Akt and p-ERK. However, it enhanced the cisplatin-induced up-regulation of p-P38 and the down-regulation of p-Akt and p-ERK in the SK-OV-3 and ES-2 OC cells (Figure 2), which induced the downstream proteins Bcl-2, Bax and active caspase-3. The elevation of Bax and active caspase-3 and of the decrease in Bcl-2 induced by cisplatin were also enhanced by the knockdown of *Zfp90* (Figure 3). Except in case of the apoptosis cascade, the migrative effect was extensively investigated.

Almost 90% of cancer deaths are attributed to cancer metastasis and damage of secondary tumors [55]. To treat OC more effectively, some analyses have focused on the inhibition of metastatic carcinoma cells and took metastasis as being crucial for cell proliferation. Qian et al., 2021, demonstrated that the kinesin family member 18A (KIF18A) was overexpressed in esophageal cancer (EC) patients, and the modulation of KIF18A could impact cancer cell migration and invasion in the EC cell lines. They also knocked-down si-KIF18A and activated the Insulin-like growth factor-II mRNA binding protein 3 (IGF2BP3) to reconfirm the role of the KIF18A function in cell movement [56]. In addition, Zhang et al., 2019, also indicated histone H3 lysine 4 (H3K4) and H3 lysine 9 (H3K9) demethylase (KDM1A) as metastasis promoters in papillary thyroid cancer. They revealed that KDM1A could increase the MMP-9 expression and activity via binding to the active site of the tissue inhibitors of metalloproteinases-1 (TIMP-1). These effects influenced the decrease in the migrative effect in the wound healing assay and invasion assay [57]. Furthermore, Si et al., 2020, expressed that the knockdown of cell adhesion molecules-1 (CADM1) influenced the growth, migration and invasion of OC cells. They also overexpressed CADM1 in OC cells and observed an increase in the cell growth and movements via a wound healing assay. The mechanism of action in CADM1 passed through the PI3K/Akt signaling pathway, which is comparable to our results [58]. We conducted a wound healing assay in SK-OV-3 and ES-2 cells to evaluate the effect of *Zfp90* knockdown on cell migration. The data indicated that the inhibition of *Zfp90* could further decrease the cisplatin-induced down-regulation of the migrative effect in both cells. The MMPs activity also expressed the same trend. The inhibition of *Zfp90* could substantially reduce the cisplatin-induced down-regulation of the MMPs activity (Figure 4). Some related protein expressions, including MMP-2, MMP-9 and E-cadherin, were also conducted. The cisplatin-induced down-regulation of MMP-2, MMP-9 and E-cadherin were greatly suppressed by the knockdown of *Zfp90* (Figure 4). Although cisplatin played a significant role in the OC treatment, the issue of drug resistance in OC still requires attention.

Drug resistance is always a critical issue in OC treatment [59,60]. Among all the related pathways, the Nrf2/HO-1 signaling pathway has been explored significantly in cisplatin sensitization [61,62,63]. Deng et al., 2020, revealed that the Nrf2/HO-1 pathway was determined as a drug resistance mechanism in SK-OV-3 cells, and the peroxisome proliferator-activated receptor-γ coactivator 1-α (PGC1α) is implicated in the regulation of Nrf2 via increasing the p-GSK3β and p-GSK3β, which would conversely modulate the transcriptional activity of PGC1α. They demonstrated that the intervention of Nrf2 or PGC1α led to the enhancement of cisplatin sensitization in SK-OV-3 and A2780 cell lines [62]. Li et al., 2021, outlined some factors that react to Nrf2 and cause resistance to cisplatin, including P62, CD99, ABCF2 and ATF2 [17]. Xia et al., 2020, indicated that the overexpression of p62 in SK-OV-3 could shield the cell against vitamin K3-induced damage via an increase in the anti-oxidant genes, such as Nrf2, and the downstream factors, including HO-1 and NQO-1 [64]. Bao et al., 2017, revealed that ABCF2 was impacted by the low expression of Nrf2, which influenced the upregulation of the cisplatin sensitization of A2780 by regulating the drug efflux pump [65]. In our studies, we propose that *Zfp90*-regulated cisplatin sensitization in SK-OV-3 and ES-2 cells might occur through the Nrf2/HO-1 pathway. The up-regulation of Nrf2 and HO-1 protein expression was significantly mitigated by the knockdown of *Zfp90*. The upstream factor BMP-7 was also conducted by an ELISA, and the up-regulation of the BMP-7 concentration was greatly inhibited by the knockdown of *Zfp90* (Figure 5). Conversely, we validate the association between *Zfp90* and the Nrf2/HO-1 pathway via treating carnosol, a facilitator of HO-1. The data expressed that the treatment of carnosol could greatly abolish the si-*Zfp90*-induced enhancement of the cytotoxic effect. The SOD activity was also evaluated, and its trend is comparable to the cell viability (Figure 6).

According to the results of our study, *Zfp90* was shown to be a potential target to deal with cisplatin chemoresistance in OC via oxidative stress and the apoptotic pathway. However, further in vivo tests should be conducted to examine its role in OC chemoresistance. The data from the GWAS also showed a great difference between OC patients and normal people in terms of *Zfp90* expression. We hope the above findings could help the development of target strategies employing *Zfp90* siRNAs to complement the conventional chemotherapies for advanced OC.

## 5. Conclusions

In conclusion, our data highlight that cisplatin treatment produces ROS that modulate apoptotic proteins expression (p-P38, p-ERK, p-Akt, Bcl-2, Bax and active caspase-3). The anti-oxidative signal (Nrf2, HO-1 and SOD) was also stimulated, which could inhibit cell migration (MMP-2, MMP-9, E-cadherin). The intervention of *Zfp90* could substantially enhance the apoptosis pathway and inhibit the migrative pathway to regulate the cisplatin sensitivity in OC cells.

## Figures and Tables

**Figure 1 cancers-15-01586-f001:**
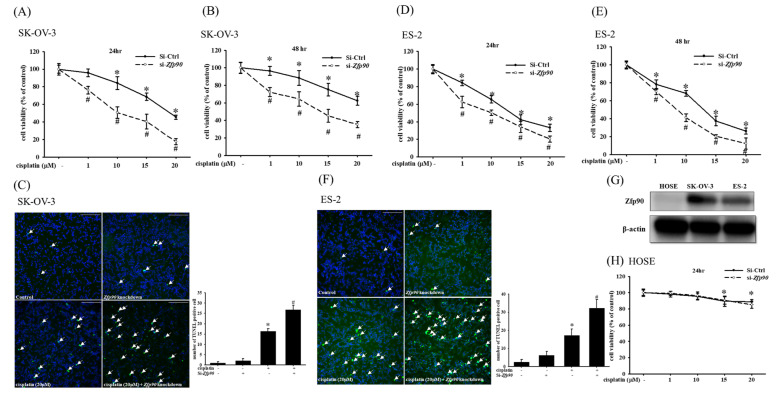
Effect of inhibiting *Zfp90* on the cisplatin sensitivity of SK−OV−3, ES−2 and human ovarian surface epithelial (HOSE) cells. SK−OV−3 and ES−2 cells were treated with various concentrations of cisplatin (1, 10, 15, 20 μM) for 24 h and 48 h in the si-Ctrl or si-*Zfp90* groups. Cell viability was determined by alamarBlue^TM^ assay. SK−OV−3 and ES−2 cells were treated with 20 μM cisplatin for 24 h in si−Control or si−*Zfp90* cells, and the apoptotic effect was identified by the TUNEL assay. Cell viability of each group at (**A**) 24 h or (**B**) 48 h were observed in SK−OV−3 cell. (**C**) TUNEL stain of each group was detected in SK−OV−3 cell. Cell viability of each group at (**D**) 24 h or (**E**) 48 h was observed in ES−2 cell. (**F**) TUNEL stain of each group was discovered in the ES−2 cell. (**G**) Zfp90 protein expression was confirmed in the human ovarian surface epithelial (HOSE) cell, SK−OV−3 and ES−2 cells by western blotting. HOSE cells were treated with different concentrations of cisplatin (1, 10, 15, 20 μM) for 24 h. (**H**) Cell viability of each group at 24 h was observed in the HOSE cell. Data are expressed as the mean ± SEM (*n* = 8) (*, *p* < 0.05, versus si−Ctrl group; #, *p* < 0.05, versus same concentration cisplatin in si−Ctrl cell, respectively).

**Figure 2 cancers-15-01586-f002:**
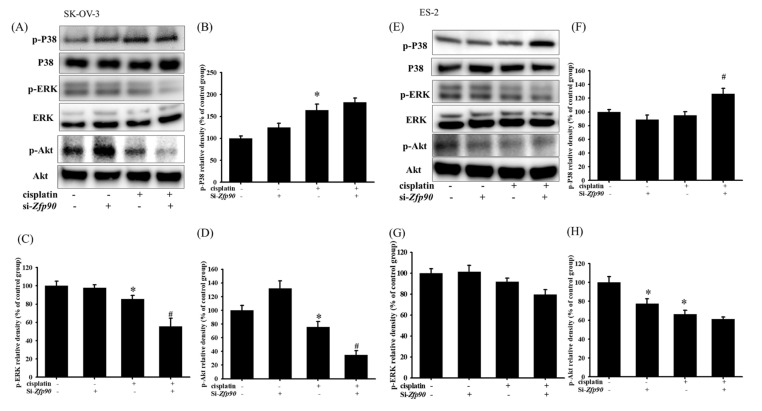
The inhibition of *Zfp90* modulated the cisplatin−induced p−P38, P38, p−ERK, ERK, p−Akt and Akt protein expression in the SK−OV−3 and ES−2 cells. The SK−OV−3 and ES−2 cells were treated with 20 μM cisplatin for 1 h in the si−Ctrl or si−*Zfp90* cells, and protein expression was identified by western blotting. (**A**) The p−P38, p−Akt and p−ERK protein expression of si−Ctrl, cisplatin, si−*Zfp90*, si−*Zfp90* and cisplatin group were detected in the SK−OV−3 cell. Quantitative results of (**B**) p−P38, (**C**) p−ERK and (**D**) p−Akt were presented. (**E**) p−P38, p−Akt and p−ERK protein expression of si−Ctrl, cisplatin, si−*Zfp90*, si−*Zfp90* and cisplatin group was also observed in the ES−2 cell. Quantitative results of (**F**) p−P38/P38, (**G**) p−ERK/ERK and (**H**) p−Akt/Akt were calculated. Total P38, ERK and Akt were used as internal control individually. The data are expressed as the mean ± SEM (n = 3) (*, *p* < 0.05, versus si−Ctrl group; #, *p* < 0.05, versus cisplatin group).

**Figure 3 cancers-15-01586-f003:**
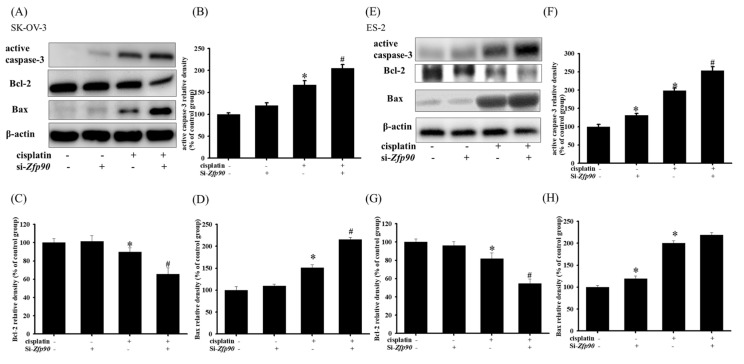
The inhibition of *Zfp90* modulated the cisplatin−induced active caspase−3, Bcl−2, Bax protein expression of SK−OV−3 and ES−2 cells. SK−OV−3 and ES−2 cells were treated with 20 μM cisplatin for 24 h in the si−Ctrl or si−*Zfp90* cells, and protein expression was determined by western blotting. (**A**) active caspase-3, Bcl−2, Bax protein expression of si−Ctrl, cisplatin, si−*Zfp90*, and the si−*Zfp90* and cisplatin group were detected in SK−OV−3 cell. Quantitative results of (**B**) active caspase−3, (**C**) Bcl−2 and (**D**) Bax were shown. (**E**) active caspase−3, Bcl−2, Bax protein expression of si−Ctrl, cisplatin, si−*Zfp90*, and the si−*Zfp90* and cisplatin group was also observed in ES−2 cell. Quantitative results of (**F**) active caspase−3, (**G**) Bcl−2, and (**H**) Bax were shown. Data are expressed as the mean ± SEM (n = 3) (*, *p* < 0.05, versus si−Ctrl group; #, *p* < 0.05, versus cisplatin group).

**Figure 4 cancers-15-01586-f004:**
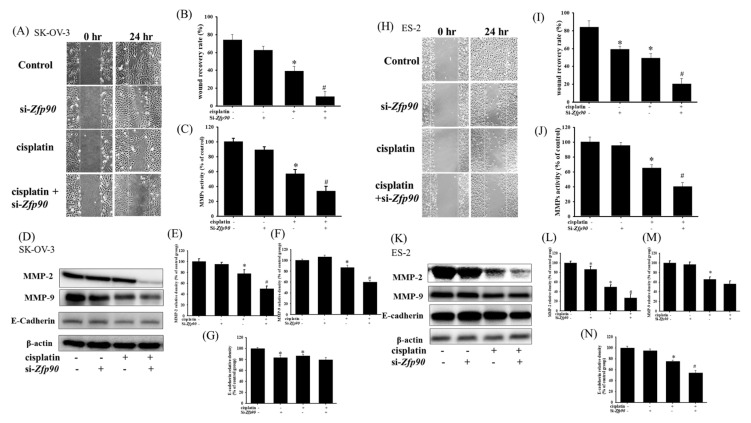
Effect of inhibiting *Zfp90* on cisplatin−induced down−regulation of the migrative effect in SK−OV−3 and ES−2 cells. SK−OV−3 and ES−2 cells were treated with 20 μM cisplatin for 24 h in the si−Control or si−*Zfp90* cells, and cell migration was identified by wound healing assay, ELISA and western blotting. (**A**) Wound healing assay of si−Ctrl, cisplatin, si−*Zfp90*, and the si−*Zfp90* and cisplatin group were observed in SK−OV−3 cell. (**B**) Quantitative results of SK−OV−3 wound healing assay were shown. (**C**) MMPs activity of each group were measured in SK−OV−3 cell. (**D**) MMP−2, MMP−9, E−cadherin protein expression of each group were also observed in SK−OV−3 cell and quantitative results of (**E**) MMP−2, (**F**) MMP−9 and (**G**) E−cadherin were shown. (**H**) Wound healing assay of si−Ctrl, cisplatin, si−*Zfp90*, and the si−*Zfp90*andcisplatin group were observed in ES−2 cell. (**I**) Quantitative results of the ES−2 wound healing assay were shown. (**J**) MMPs activity of each group were measured in ES−2 cell. (**K**) MMP−2, MMP−9, E−cadherin protein expression of each group were observed in SK−OV−3 cell and quantitative results of (**L**) MMP−2, (**M**) MMP−9, and (**N**) E−cadherin were shown. Data are expressed as the mean ± SEM (*n* = 3) (*, *p* < 0.05, versus si−Ctrl group; #, *p* < 0.05, versus cisplatin group).

**Figure 5 cancers-15-01586-f005:**
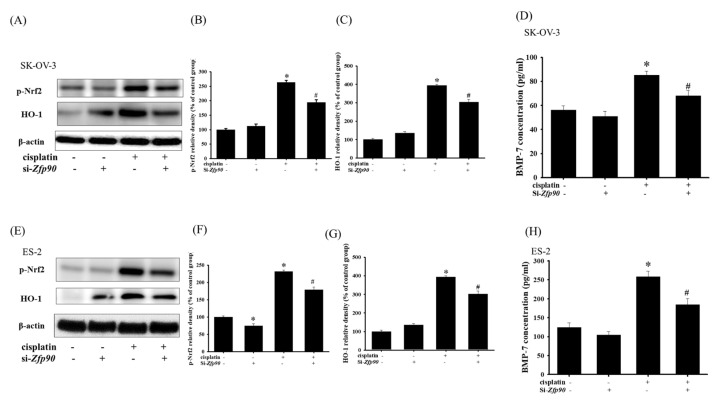
Effect of inhibiting *Zfp90* on cisplatin−induced up−regulation of p−Nrf2, HO−1 protein expression and BMP−7 protein secretion in SK−OV−3, and ES−2 cells. SK−OV−3 and ES−2 cells were treated with 20 μM cisplatin for 24 h in the si−Ctrl or si−*Zfp90* cells, and protein expression was ascertained by western blotting. (**A**) p−Nrf2, HO−1 protein expression of si−Ctrl, cisplatin, si−*Zfp90*, and the si−*Zfp90*andcisplatin group were observed in SK−OV−3 cell. Quantitative results of (**B**) p−Nrf2, and (**C**) HO−1 were shown. SK−OV−3 and ES−2 cells were treated with 20 μM cisplatin for 24 h in the si−Ctrl or si−*Zfp90* cells and BMP−7 protein secretion was determined by ELISA. (**D**) BMP−7 protein secretion of each group were observed in SK−OV−3 cell. (**E**) p−Nrf2, and HO−1 protein expression of si−Ctrl, cisplatin, si−*Zfp90*, and the si−*Zfp90*andcisplatin group were also observed in ES−2 cell. Quantitative results of (**F**) p−Nrf2, and (**G**) HO−1 were detected. (**H**) BMP−7 protein secretion of each group were also observed in ES−2 cell. Data are presented as the mean ± SEM (n = 3) (*, *p* < 0.05, versus si−Ctrl group; #, *p* < 0.05, versus cisplatin group).

**Figure 6 cancers-15-01586-f006:**
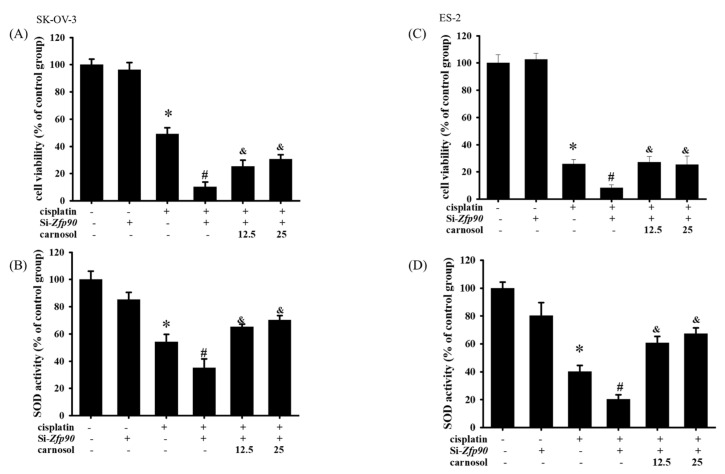
Effect of stimulation HO−1 on si−Zfp90−induced enhancement of cell death and down−regulation of SOD activity in SK−OV−3 and ES−2 cells. SK−OV−3 and ES−2 cells were pre−treated with carnosol (12.5 or 25 μM) for 1 h followed by 20 μM cisplatin for 24 h in the si−*Zfp90* cells. Cell viability was determined by alamarBlue^TM^ assay and superoxide dismutase (SOD) activity was determined by Superoxide Dismutase Activity Assay Kit (Colorimetric). (**A**) Cell viability of si−Ctrl, cisplatin, si−*Zfp90*, si−*Zfp90* and cisplatin, si−*Zfp90* and cisplatin and 12.5 μM carnosol and si−*Zfp90* and cisplatin and 25 μM carnosol at 24 h were detected in SK−OV−3 cell. (**B**) SOD activity of si−Ctrl, cisplatin, si−*Zfp90*, si−*Zfp90* and cisplatin, si−*Zfp90* and cisplatin and 12.5 μM carnosol and si−*Zfp90* and cisplatin and 25 μM carnosol at 24 h were also observed in SK−OV−3 cell. (**C**) Cell viability of each group at 24 h were also observed in ES−2 cell. (**D**) SOD activity of each group were observed in ES−2 cell. Data are presented as the mean ± SEM (*n* = 8) (*, *p* < 0.05, versus si−Ctrl group; #, *p* < 0.05, versus cisplatin group; and, &, *p* < 0.05, versus si−*Zfp90* plus cisplatin group).

## Data Availability

All relevant data are within the manuscript.

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
