# Peer review of "Zinc Finger Protein 90 Knockdown Promotes Cisplatin Sensitivity via Nrf2/HO-1 Pathway in Ovarian Cancer Cell"

_cancers, 2023, doi:10.3390/cancers15051586_

Round 1

Reviewer 1 Report

The paper detected the co-factor treatment, cisplatin, and Zfp90, impact the apoptosis pathway by employing siRNA to knock down Zfp90 and measure the apoptotic protein expression, and-oxidative signal proteins, and cell migration-related protein abundance after treating with 20um cisplatin. Moreover, the authors tested the hypothesis on two types of cells, SK-OV-3 cell and ES-2 cell.

The paper is well written. However, for different detection methods, the authors used different incubation times for cisplatin. For example, for apoptotic study, the author wrote "treatment 8h with Cisplatin", but when studying cisplatin-modulated anti-migrative effect, the author wrote "treatment 24h with cisplatin'. At the same time, in the main part and the legend part, the expression of the cisplatin-treatment time is confusing. For example: Page 5, line 233, the author describes the treatment of cells for 1h, but in the corresponding legend (page 6, line 248) the author writes "treatment of cells for 2h"; the same problem, page 6, line 259, the author mentions the cells were processed for 8 hours, but on Page7, line 275, it was written that the cells were processed for 24 hours. This confusing expression greatly reduces the credibility of the data in the article.

Some other major comments:

1.In figure 1, the author a showed that 10, 15, and 20μM cisplatin could significantly inhibit SK-OV-3  and ES-2 cell viability are similar. However, the paper (PMID: 25973302) demonstrated that cells exhibited different IC50 values with cisplatin, which  show inconsistent with the author's in this paper. Please double-check the results.

2.On page 5, line 230. The author mentioned that 20μM cisplatin significantly dysregulated some phosphorylation protein, such as p-P38, p-ERK, and p-Akt. Such representations are inaccurate. Because all the antibodies utilized in this paper are against the phosphorylation changes of specific- site. Changes in one phosphorylation site do not reflect changes in overall protein levels. Do the authors want to see changes in the overall protein or changes in a particular phosphorylation site? If the latter, the author needs to provide the corresponding protein expression to show that the change of phosphorylation modification is not due to the change of the overall protein expression.

Some minor comments:

1.       Page4, line 147, 10%SDS duplicated.

2.       Page 5, figure 1 legend D and E in line 224, missing the ES-2 cell. It should be” Cell viability of each 223 group at (D) 24 hr or (E) 48 hr were observed in ES-2 cell”.

Author Response

Reviewer 1.

The paper detected the co-factor treatment, cisplatin, and Zfp90, impact the apoptosis pathway by employing siRNA to knock down Zfp90 and measure the apoptotic protein expression, and-oxidative signal proteins, and cell migration-related protein abundance after treating with 20um cisplatin. Moreover, the authors tested the hypothesis on two types of cells, SK-OV-3 cell and ES-2 cell.

Q1. The paper is well written. However, for different detection methods, the authors used different incubation times for cisplatin. For example, for apoptotic study, the author wrote "treatment 8h with Cisplatin", but when studying cisplatin-modulated anti-migrative effect, the author wrote "treatment 24h with cisplatin'. At the same time, in the main part and the legend part, the expression of the cisplatin-treatment time is confusing. For example: Page 5, line 233, the author describes the treatment of cells for 1h, but in the corresponding legend (page 6, line 248) the author writes "treatment of cells for 2h"; the same problem, page 6, line 259, the author mentions the cells were processed for 8 hours, but on Page7, line 275, it was written that the cells were processed for 24 hours. This confusing expression greatly reduces the credibility of the data in the article.

Our reply: Thanks for your careful inspection, we investigated the anti-migrative effect (MMP9, MMP-2, E-cadherin) and apoptotic effect (active caspase-3, bcl-2, Bax) after cisplatin treatment for 24 hr in SK-OV-3 or ES-2 cell line as referred literature [1,2]. The PI3K/Akt, ERK and p38 protein were all performed after 1 hr treatment of cisplatin. We apologized for our carelessness and corrected the incubation method as followed

SK-OV-3 and ES-2 cells were treated with 20μM cisplatin for 1 hr in the si-Ctrl or si-Zfp90 cells, and protein expression was determined by western blotting” in Fig 2 legend; “We then examined phosphorylation of p-P38, p-ERK, and p-Akt and downstream apoptotic related protein expression to further confirm the relationship between Zfp90 and cisplatin sensitization. The protein expression of 20μM cisplatin in SK-OV-3 and ES-2 cell with or without si-Zfp90 were determined by western blotting after 1 hr treatment.” in Result section.

SK-OV-3 and ES-2 cells were treated with 20μM cisplatin for 24 hr in the si-Ctrl or si-Zfp90 cells, and protein expression was determined by western blotting.” in Fig 3 legend; “The apoptotic protein of 20μM cisplatin in SK-OV-3 and ES-2 cell with or without si-Zfp90 were determined by western blotting after 24 hr treatment. The data showed that 20μM cisplatin significantly increased active-caspase-3, Bax and decreased Bcl-2 protein expression (*, p<0.05, versus si-Ctrl group).” in Result section.

  1. Chen, S.; Li, Z.; Wang, Y.; Fan, S. BTN3A3 inhibits the proliferation, migration and invasion of ovarian cancer cells by regulating ERK1/2 phosphorylation. 2022, 12, doi:10.3389/fonc.2022.952425.
  2. Chen, Q.; Zhang, J.; He, Y.; Wang, Y. hsa_circ_0061140 Knockdown Reverses FOXM1-Mediated Cell Growth and Metastasis in Ovarian Cancer through miR-370 Sponge Activity. Molecular Therapy - Nucleic Acids 2018, 13, 55-63, doi:https://doi.org/10.1016/j.omtn.2018.08.010.

Some other major comments:

Q2. In figure 1, the author a showed that 10, 15, and 20μM cisplatin could significantly inhibit SK-OV-3 and ES-2 cell viability are similar. However, the paper (PMID: 25973302) demonstrated that cells exhibited different IC50 values with cisplatin, which show inconsistent with the author's in this paper. Please double-check the results.

Our reply: Thanks for your careful review, we’ve referred to the conditions of the paper (PMID:25973302) [1] and changed the implanted cell number to 2 x 103 of ES-2, SK-OV-3 cell and showed the data as followed

Figure 1. Effect of inhibiting Zfp90 on cisplatin sensitivity of SK-OV-3, ES-2 and human ovarian surface epithelial (HOSE) cells.

SK-OV-3 and ES-2 cells were treated with different concentrations of cisplatin (1, 10, 15, 20μM) for 24 hr and 48 hr in the si-Ctrl or si-Zfp90 groups. Cell viability was determined by alamarBlueTM assay. SK-OV-3 and ES-2 cells were treated with 20μM cisplatin for 24 hr in si-Control or si-Zfp90 cells, and the apoptotic effect was determined by TUNEL assay. Cell viability of each group at (A) 24 hr or (B) 48 hr were observed in SK-OV-3 cell. (C) TUNEL stain of each group were observed in SK-OV-3 cell. Cell viability of each group at (D) 24 hr or (E) 48 hr were observed in ES-2 cell. (F) TUNEL stain of each group were observed in ES-2 cell. (G) Zfp90 protein expression was confirmed in human ovarian surface epithelial (HOSE) cell, SK-OV-3 and ES-2 cells by western blotting. HOSE cells were treated with different concentrations of cisplatin (1, 10, 15, 20μM) for 24 hr. (H) Cell viability of each group at 24 hr were observed in HOSE cell. Data are presented as the mean±SEM (n=8) (*, p<0.05, versus si-Ctrl group; #, p<0.05, versus same concentration cisplatin in si-Ctrl cell respectively).

The revised data showed a similar trend as the paper (PMID:25973302).

  1. Wu, X.; Zhi, X.; Ji, M.; Wang, Q.; Li, Y.; Xie, J.; Zhao, S. Midkine as a potential diagnostic marker in epithelial ovarian cancer for cisplatin/paclitaxel combination clinical therapy. American journal of cancer research 2015, 5, 629-638.

Q3. 2.On page 5, line 230. The author mentioned that 20μM cisplatin significantly dysregulated some phosphorylation protein, such as p-P38, p-ERK, and p-Akt. Such representations are inaccurate. Because all the antibodies utilized in this paper are against the phosphorylation changes of specific- site. Changes in one phosphorylation site do not reflect changes in overall protein levels. Do the authors want to see changes in the overall protein or changes in a particular phosphorylation site? If the latter, the author needs to provide the corresponding protein expression to show that the change of phosphorylation modification is not due to the change of the overall protein expression.

Our reply: Thanks for your valuable suggestion, we’ve added the total P38, ERK and Akt in Figure 2 as followed

Figure 2. The inhibition of Zfp90 modulated the cisplatin-induced p-P38, p-ERK, p-Akt protein expression in SK-OV-3 and ES-2 cells.

SK-OV-3 and ES-2 cells were treated with 20μM cisplatin for 1 hr in the si-Ctrl or si-Zfp90 cells, and protein expression was determined by western blotting. (A) p-P38, p-Akt and p-ERK protein expression of si-Ctrl, cisplatin, si-Zfp90, si-Zfp90+cisplatin group were observed in SK-OV-3 cell. Quantitative results of (B) p-P38, (C) p-ERK and (D) p-Akt were shown. (E) p-P38, p-Akt and p-ERK protein expression of si-Ctrl, cisplatin, si-Zfp90, si-Zfp90+cisplatin group were also observed in ES-2 cell. Quantitative results of (F) p-P38, (G) p-ERK and (H) p-Akt were measured. Total P38, ERK and Akt were used as internal control individually. Data are presented as the mean±SEM (n=3) (*, p<0.05, versus si-Ctrl group; #, p<0.05, versus cisplatin group).

Some minor comments:

Q3. Page4, line 147, 10%SDS duplicated.

Our reply: Thanks for the careful review, we’ve deleted the duplicate words.

  1. Page 5, figure 1 legend D and E in line 224, missing the ES-2 cell. It should be” Cell viability of each 223 group at (D) 24 hr or (E) 48 hr were observed in ES-2 cell”.

Our reply: Thanks for the careful reminder, we’ve corrected the sentences as followed “Cell viability of each groups at (D) 24 hr or (E) 48 hr were observed in ES-2 cell”.

Reviewer 2 Report

Comments and Suggestions for the Authors.

MS ID: cancers- 2238850

Article.

Title: Zinc finger protein 90 knockdown promotes cisplatin sensitivity via Nrf2/HO-1 pathway in ovarian cancer cell.

Authors: Ching-Hu Wu et al.

The manuscript presented by Ching-Hu Wu et al "Zinc finger protein 90 knockdown promotes cisplatin sensitivity via Nrf2/HO-1 pathway in ovarian cancer cell" is very poorly written presented and described. The first figure itself is very hard to get it and understand what the authors are saying as the annotations and legends are not at all visible. Similarly rest of the figures and most of the data is very observational and descriptive to support the authors claim.

The language is very hard to understand and needs extensive upgrade.

Overall the data presented and the way it is written is not up to the journals standards and hence it should be not consider for publish in Cancers.

Author Response

Reviewer 2

In this manuscript, the authors showed that the inhibition of Zfp90 could sensitize ovarian cancer cell lines to cisplatin. Mechanismly, they found that the knockdown of Zfp90 could decrease p-ERK, p-AKT and activate apoptosis pathway.

The authors should address following questions to strengthen the manuscript.

Q1. The authors should knockdown Zfp90 in ovarian normal cell lines and measure the cell proliferation and migration with or without cisplatin to prove that this effect is specific in cancer cells but not in normal cells.

Our reply: Thanks for your valuable suggestion, we’ve added some data om the results section about human ovarian surface epithelial cells as followed

We used the human ovarian surface epithelial cells (HOSE) as normal cell compare with two OC cell line. Zfp90 protein expression was performed in HOSE, SK-OV-3 and ES-2 cell (Fig 1G). The cytotoxic effect of different concentration (1, 10, 15 and 20μM) of cisplatin in HOSE cell with or without si-Zfp90 were performed at 24 hr (Fig 1H). The data showed that inhibition of Zfp90 didn’t affect cisplatin-induced damage to HOSE cell.

and added a paragraph in discussion as followed

Besides, we also confirmed the effect of knockdown of Zfp90 in normal human ovarian surface epithelial (HOSE) cell didn’t cause a significant difference in cell viability (Fig 1H). HOSE cell was used as in normal cells compared to the OC cell line in previous studies [1,2]. Xie et al., 2016 used human normal ovarian surface epithelial cell to compare the expression of MUS81 with SOC tissues at both transcript and protein levels, and the expression level of MUS81 protein in ovarian cancer cell lines was also higher than that in human normal ovarian surface epithelial cell line. The above result showed a simliar trend as us. The low expression of Zfp90 might also explained the minimally effect on the cell viability by knockdown of Zfp90.

  1. Jiang, W.; Cong, Q.; Wang, Y.; Ye, B.; Xu, C.J.I.C.T. Ginkgo may sensitize ovarian cancer cells to cisplatin: antiproliferative and apoptosis-inducing effects of ginkgolide B on ovarian cancer cells. 2014, 13, NP10-NP17.
  2. Xie, S.; Zheng, H.; Wen, X.; Sun, J.; Wang, Y.; Gao, X.; Guo, L.; Lu, R. MUS81 is associated with cell proliferation and cisplatin sensitivity in serous ovarian cancer. Biochemical and Biophysical Research Communications 2016, 476, 493-500, doi:https://doi.org/10.1016/j.bbrc.2016.05.152.

Q2.  The authors should use CRISPR or shRNA to generate stable knockout or knockdown ovarian cancer cells and perform xenograft model in combination with cisplatin to prove that the tumor growth and metastasis is decreased in vivo.

Our reply: We have to apologize for the lack of stable knockout cancer cells or xenograft data. We’ll try to make it complete in our future study. Besides, our study originated on the analysis of the genome-wide association studies (GWAS) data which is aimed at identifying associations of genotypes with phenotypes by testing for differences in the allele frequency of genetic variants between individuals who are ancestrally similar but differ phenotypically. We suggest our current finding might support further investigation on the role of Zfp90 in OC chemoresistance.

Reviewer 3 Report

In this manuscript, the authors showed that the inhibition of Zfp90 could sensitize ovarian cancer cell lines to cisplatin. Mechanismly, they found that the knockdown of Zfp90 could decrease p-ERK, p-AKT and activate apoptosis pathway.

The authors should address following questions to strengthen the manuscript.

1.  The authors should knockdown Zfp90 in ovarian normal cell lines and measure the cell proliferation and migration with or without cisplatin to prove that this effect is specific in cancer cells but not in normal cells.

2.  The authors should use CRISPR or shRNA to generate stable knockout or knockdown ovarian cancer cells and perform xenograft model in combination with cisplatin to prove that the tumor growth and metastasis is decreased in vivo.

Author Response

Title: Zinc finger protein 90 knockdown promotes cisplatin sensitivity via Nrf2/HO-1

pathway in ovarian cancer cell.

Authors: Ching-Hu Wu et al.

The manuscript presented by Ching-Hu Wu et al "Zinc finger protein 90 knockdown promotes cisplatin sensitivity via Nrf2/HO-1 pathway in ovarian cancer cell" is very poorly written presented and described. The first figure itself is very hard to get it and understand what the authors are saying as the annotations and legends are not at all visible. Similarly rest of the figures and most of the data is very observational and descriptive to support the authors claim. The language is very hard to understand and needs extensive upgrade. Over all the data presented and the way it is written is not up to the journals standards and hence it should be not consider for publish in Cancers.

Our reply: Thank you for your valuable and thoughtful comments. We have carefully checked and improved the English writing in the revised manuscript. We also supplement some data in the result. We hope the revised manuscript could meet journal’s criteria.

Round 2

Reviewer 1 Report

The author answered the review comments very well and agreed to receive

Reviewer 2 Report

Over all the authors has significantly improved the manuscript and followed the changes required by the authors. Their reason for missing data is justifiable. English syntax errors and grammar also significantly improved. Hence it can be recommended for the publication.

Good Luck!.